# Label-free analysis of physiological hyaluronan size distribution with a solid-state nanopore sensor

Felipe Rivas[1], Osama K. Zahid[1], Heidi L. Reesink[2], Bridgette T. Peal[2], Alan J. Nixon[2], Paul L. DeAngelis[3], Aleksander Skardal [1,4,5], Elaheh Rahbar[1] & Adam R. Hall[1,4,5]

Hyaluronan (or hyaluronic acid, HA) is a ubiquitous molecule that plays critical roles in numerous physiological functions in vivo, including tissue hydration, inflammation, and joint lubrication. Both the abundance and size distribution of HA in biological fluids are recognized as robust indicators of various pathologies and disease progressions. However, such analyses remain challenging because conventional methods are not sufficiently sensitive, have limited dynamic range, and/or are only semi-quantitative. Here we demonstrate label-free detection and molecular weight discrimination of HA with a solid-state nanopore sensor. We first employ synthetic HA polymers to validate the measurement approach and then use the platform to determine the size distribution of as little as 10 ng of HA extracted directly from synovial fluid in an equine model of osteoarthritis. Our results establish a quantitative method for assessment of a significant molecular biomarker that bridges a gap in the current state of the art.

[1] Virginia Tech-Wake Forest University School of Biomedical Engineering and Sciences, Wake Forest School of Medicine, Winston-Salem, NC 27101, USA. [2] Department of Clinical Sciences, College of Veterinary Medicine, Cornell University, Ithaca, NY 14853, USA. [3] Department of Biochemistry and Molecular Biology, University of Oklahoma Health Sciences Center, Oklahoma, OK 73104, USA. [4] Institute for Regenerative Medicine, Wake Forest School of Medicine, Winston-Salem, NC 27101, USA. [5] Comprehensive Cancer Center, Wake Forest School of Medicine, Winston-Salem, NC 27157, USA. Correspondence and requests for materials should be addressed to E.R. (email: erahbar@wakehealth.edu) or to A.R.H. (email: arhall@wakehealth.edu)

Hyaluronan (or hyaluronic acid, HA)[1] is a polyanionic linear chain in the glycosaminoglycan (GAG) family featuring the alternating disaccharide repeat structure [-4-D-glucuronic acid-β1-3-N-acetylglucosamine-β1-]$_n$. Distributed widely throughout mammalian cells and tissues, the biomechanical and biochemical properties of HA support its involvement in myriad physiological functions, including hydration and turgidity maintenance of tissue[1], extracellular matrix structure[1], regulation of innate immunity[2], and protection and lubrication of joints[3]. As a result of this versatility, HA is considered a promising bioindicator of pathophysiology and inflammation, and has consequently been targeted for disease-specific diagnostics[4,5]. While the molecular weight (MW) of naturally occurring HA is typically[6] in the range of $10^5$–$10^7$ Da (~250–25,000 disaccharide units, each ~1 nm in length), its size within this range is a critical determinant of the molecule's function in vivo. For example, high-MW HA (>1000 kDa) is highly viscous and appears to display anti-inflammatory and immunosuppressive properties[7]; whereas, low-MW HA (generally <500 kDa) can induce the release of pro-inflammatory cytokines from macrophages[8,9]. Furthermore, high-MW HA is far more responsible than low-MW HA for the lubricating properties of synovial fluid. Consequently, both the abundance and size distribution of HA are important biomarkers for disease pathologies and are essential to

understanding the immunomodulatory and joint lubrication roles of HA in vivo.

Unfortunately, current technologies for HA detection and size differentiation have significant limitations. For example, techniques similar to enzyme-linked immunosorbent assays[10,11] (ELISA) can be used for quantitative detection of HA, but have limited capacity to differentiate by MW and in some formats can neglect low-MW species. Supplementation of the approach with fractionation methods, such as size exclusion chromatography (SEC) enables discrete size ranges of HA to be quantified, but the nature of SEC (including slow column flow rates and long run times) places practical constraints on the number of fractions and samples that can be examined. Conversely, multi-angle laser light scattering (MALLS[12]) can report on HA MW, but is not intrinsically quantitative, has limited precision, and is relatively insensitive to low-MW fragments. Mass spectrometry (MS)[13–15] is also capable of resolving MW differences, but in addition to requiring expensive and complex instrumentation, cannot probe HA larger than ~100 kDa. As a result of these considerations, the most widely used HA assessment approach is agarose or polyacrylamide gel electrophoresis[16,17], through which band intensity and position can be analyzed to denote a size distribution. However, this method is slow, requires large sample size (fluid volume and HA mass), requires calibrated standards (e.g.,

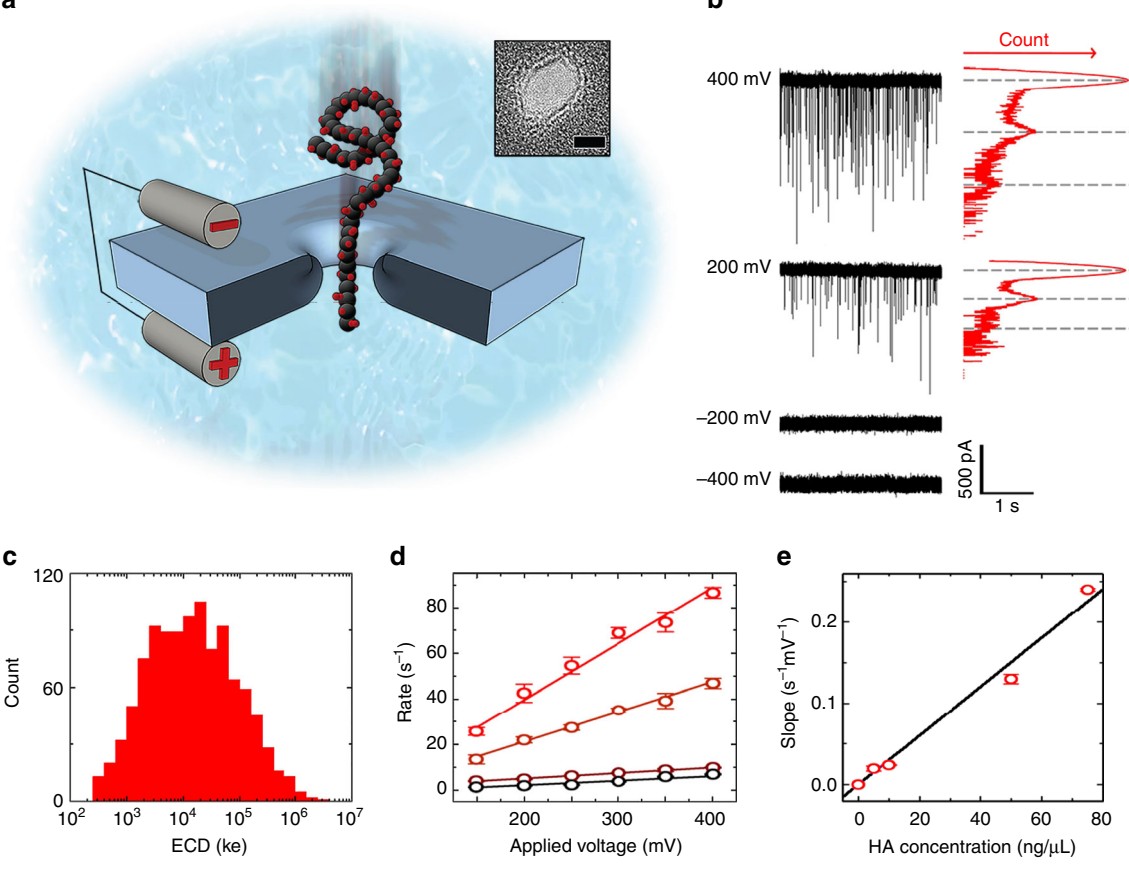

**Fig. 1** SS-nanopore detection of polydisperse HA. **a** Schematic representation of electrophoretic translocation of HA through a SS-nanopore. Inset: transmission electromicrograph of a typical SS-nanopore fabricated with the same procedure used here. Scale bar, 5 nm. **b** Raw current traces obtained from a 6.5 nm SS-nanopore with HA introduced on one chamber (cis-) and indicated voltage applied to the other (trans-). Events were observed only toward positive bias. All-points histograms (red) show quantized current levels (dashed lines), indicating molecular folding. **c** Typical ECD histogram for polydisperse HA (n = 1067) measured at 200 mV. **d** Voltage-dependent event rate for three concentrations of polydisperse HA (t-b: 75, 50, 10, and 5 ng/μl). N-values (number of uninterrupted current traces) are listed in Supplementary Table 3 and error bars are standard deviations. Solid lines are linear fits to the data points. **e** Slopes from **d** showing a linear dependence (solid line) on net HA concentration. Error bars are errors of the fits

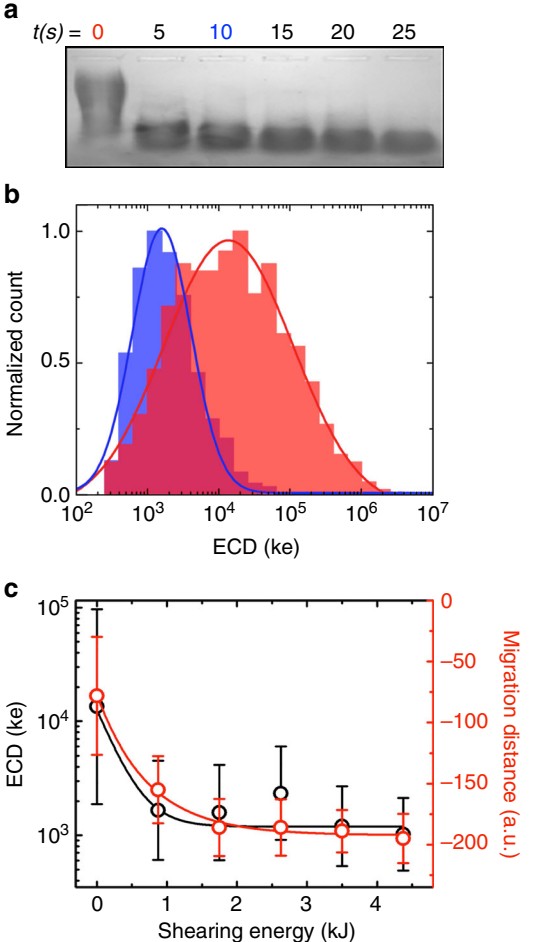

**Fig. 2** Assessment of HA mechanical shearing. **a** Gel image of polydisperse HA under increasing durations of exposure to a 175 W ultrasonic shearing bath, demonstrating increasing fragmentation. **b** ECD histograms obtained from the 0 s (red) and 10 s (blue) samples from **a**. Solid lines are log-normal fits (Gaussian on a log scale) to the data. We observe a shift in the mean from $1.3 \times 10^4$ ke (red) to $1.6 \times 10^3$ ke (blue) after shearing with population width (standard deviation) reduced from $9.4 \times 10^4$ ke (red) to $3.5 \times 10^3$ ke (blue). **c** HA ECD (left, black) and gel migration distance (right, red) for all investigated shearing conditions, yielding nearly identical trends. Solid lines are exponential fits to the respective data. In **b**, **c** ECD data consist of $t = 0$ ($n = 1067$, same data as Fig. 1c), 5 ($n = 6331$), 10 ($n = 1247$), 15 ($n = 617$), 20 ($n = 1008$), and 25 s ($n = 781$) and error bars represent the standard deviations of log-normal fits to the datasets

measured via MALLS or MS), and provides only semi-quantitative data.

Solid-state (SS-) nanopores[18,19] are an emerging platform for sensitive molecular analysis. The system uses a nanometer-scale aperture in a thin membrane (Fig. 1a, inset), positioned as the only fluid connection between two reservoirs of an electrolyte solution. An applied voltage is used to generate a strong electric field inside the opening that impels charged molecules electrophoretically through the pore and into the opposing chamber (Fig. 1a). During their residence inside the nanopore, each molecule occupies space that would otherwise be occupied by ions contributing to the electrical signal, and so their passage is marked by a temporary reduction (an "event") in the measured transmembrane ionic current. Analysis of the amplitude and the duration of events typically reports on the diameter and the contour length of the translocating molecules, respectively. The concept of resistive pulse sensing was first applied to HA by

Fennouri, et al.[20,21], using the aerolysin protein pore. However, the dynamic range of that system is very narrow; direct assessment was limited to small HA (<10 individual sugar residues) with only indirect evidence of larger molecule detection.

Here we establish the utility of fabricated SS-nanopores as a quantitative analytical tool for assessing HA. We first use synthetic HA to demonstrate that polysaccharides can be probed directly with the platform and to suggest the size-dependent nature of the measurement approach. We then employ HA populations with narrow size distributions to show that MW can be determined on a per molecule basis from the translocation signal. Finally, we demonstrate that our SS-nanopore approach can report on the size distribution of physiological HA isolated from the synovial fluid of an equine model of osteoarthritis (OA). The flexibility of our platform enables both detection and MW discrimination across a broad range of molecular sizes and its speed and quantitative output indicate a direct route to translational applications.

## Results

**SS-nanopore measurement of polydisperse HA.** As an initial assessment of the utility of SS-nanopores to probe HA, we first conducted a set of experiments using a polydisperse (i.e., broad MW distribution) mixture of HA isolated from *Streptococcus zooepidemicus* fermentation (Methods section). The resulting current traces (Fig. 1b) confirmed the ability of SS-nanopores to resolve HA easily, typically yielding events that were at least five standard deviations ($5\sigma$) above the noise floor. As a negatively charged molecule (surface charge density of $-0.32\ \mathrm{C/m^2}$, with a low-isoelectric point[22] of 2.5), HA was observed to move only toward the positive bias, indicating that its translocation was governed predominantly by electrophoresis[23]. Additionally, by reversing the applied bias after a measurement, we measured recaptured HA events (Supplementary Figure 1), confirming that the molecules fully translocated through the pore[24]. Turning to event characteristics, we noted integral variation in the measured translocation event depth histograms (Fig. 1b, right) that were suggestive of stochastic variations in molecular folding conformation during threading (Supplementary Figure 2), similar to past measurements with DNA[25,26]. While event durations have typically been more correlated with MW than depth in previous reports[27], signal variations of this kind could skew the data, since folded molecules translocate more rapidly than unfolded ones. Consequently, we utilized for our analyses the experimental factor of event charge deficit (ECD)[28], or integrated area defined by each event, such that a lower ECD corresponds to a lower MW HA chain. We chose this value because it comprises both event amplitude and duration, and thereby normalized potential differences in molecular conformation. Considering only event duration (Supplementary Figure 3), a folded molecule would appear smaller than an unfolded molecule of the same length. For our data, a typical polydisperse HA ECD histogram (Fig. 1c) showed a log-normal distribution (i.e., Gaussian on a log scale) spanning over four orders of magnitude. This wide population was indicative of the broad MW distribution within the sample.

Further probing the translocation dynamics of polydisperse HA through SS-nanopores, we measured the dependence of molecular capture rate on both applied voltage and net sample concentration (Fig. 1d). For all measured conditions, we observed a linear relationship between voltage and event rate, indicating a diffusion-limited translocation regime[29] and suggesting that there was no significant energetic barrier related to entry of HA into the confined space of the nanopore for our system[30]. Crucially, another consequence of diffusion-limited kinetics is an absence of size dependence in event rate[29], enabling an unbiased

 3

representation of MW distribution in the SS-nanopore signal. We also observed that event rates were strongly impacted by the net concentration of polydisperse HA in solution. Measurements yielded a linear response in recorded event rate dependence (slope) between 5 and 75 ng/µl (Fig. 1e). Featuring an intercept at 0, this result suggested that arbitrarily low concentrations could in principle be probed with a concomitant reduction in measured event rate. Furthermore, translocations could also be detected above 75 ng/µl, but often caused clogging at high-applied voltages, and therefore were not included here. Taken as a whole, this predictable variation indicated a route toward direct quantification of total HA with SS-nanopores, similar to previous studies on nucleic acids[31] and nucleoprotein-protein complexes[32].

A critical objective of our analysis was MW discrimination. As an initial test to demonstrate the ability of SS-nanopores to resolve differences in HA size, we first used ultrasonic shearing to fragment the same polydisperse material artificially. Separate aliquots of polydisperse HA were mechanically sheared using constant ultrasonication energy across a range of time durations, such that treated HA chains would be reduced in size to increasingly smaller chain lengths. The samples were first examined by agarose gel electrophoresis (Fig. 2a), showing both a reduced population width and a greater migration distance as shearing power was increased, thereby indicating narrowing distributions with smaller mean MW. This material was subsequently measured by the SS-nanopore. ECD distributions (Fig. 2b and Supplementary Figure 4) for the untreated control and a representative sheared sample ($t = 10$ s, corresponding to ~1.8 kJ shearing energy) agreed qualitatively with gel observations, showing a narrower distribution and a clear shift toward lower ECD. Indeed, a comparison of ECD distributions (Supplementary Table 1) with image analysis of the gel across shearing conditions demonstrated remarkable agreement between the two independent datasets (Fig. 2c), and suggested a straightforward correlation between HA MW and measured ECD from the SS-nanopore.

**MW discrimination with quasi-monodisperse HA**. Having demonstrated HA detection with SS-nanopores with an initial validation of the size dependence of the approach, we next pursued direct MW discrimination by examining quasi-monodisperse (i.e., very narrow size distribution approaching the ideal of a single MW) HA. For these studies, discrete samples of HA ranging in MW from 54 kDa to 2.4 MDa were produced via an established synthetic polymerization method[33] yielding HA typically within ±5% of mean MW, as confirmed by gel electrophoresis (Fig. 3a). Lane intensity analysis (Fig. 3b) showed discrete populations for the set of quasi-monodisperse HA, demonstrating the experimental precision achievable by gel. Similarly, we observed a series of defined peaks in the measured ECD (Fig. 3c) upon probing the same materials individually by SS-nanopore. The population for each quasi-monodisperse peak was considerably narrower than that measured for polydisperse HA (c.f. Figure 1c). Indeed, this narrowness suggested a higher resolution for the nanopore sensor than for gel analysis. We found that ECD peak separations reduced for lower MW samples, but were distinguishable down to ~80 kDa under our conditions. For the largest samples (1.1 and 2.4 MDas), we also observed some low ECD background signal that we attributed to fragmentation during handling or storage. Notably, a similar background was also visible on gel in the form of a smear in those two lanes (Fig. 3a), further supporting the validity of our measurements.

Plotting the mean ECD for all quasi-monodisperse HA samples, we found regular variation with respect to MW across

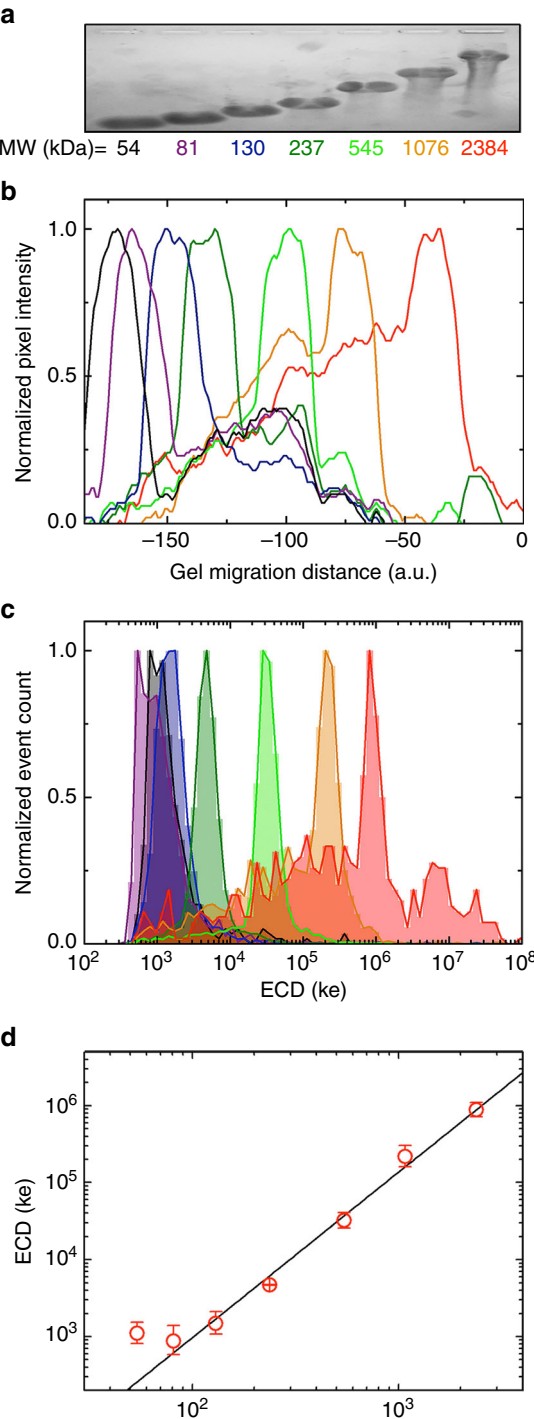

**Fig. 3** SS-nanopore analysis of quasi-monodisperse HA. **a** Gel image of quasi-monodisperse HA samples. **b** Normalized electropherogram of gel image intensity for each MW sample. Colors match MW labels used in **a**. **c** ECD histograms for each MW sample measured at 200 mV applied voltage, with number of events considered: 54 ($n = 344$), 81 ($n = 1031$), 130 ($n = 3667$), 237 ($n = 7835$), 545 ($n = 5012$), 1076 ($n = 1743$), and 2384 kDa ($n = 640$). Colors match MW labels used in **a**. **d** Relationship between ECD measured by SS-nanopore and HA MW from **c**. Error bars represent the standard deviations of log-normal fits to the datasets. Solid line is a power law fit ($\alpha = 2.23$) to the data down to 81 kDa (see Supplementary Table 1 for fit details)

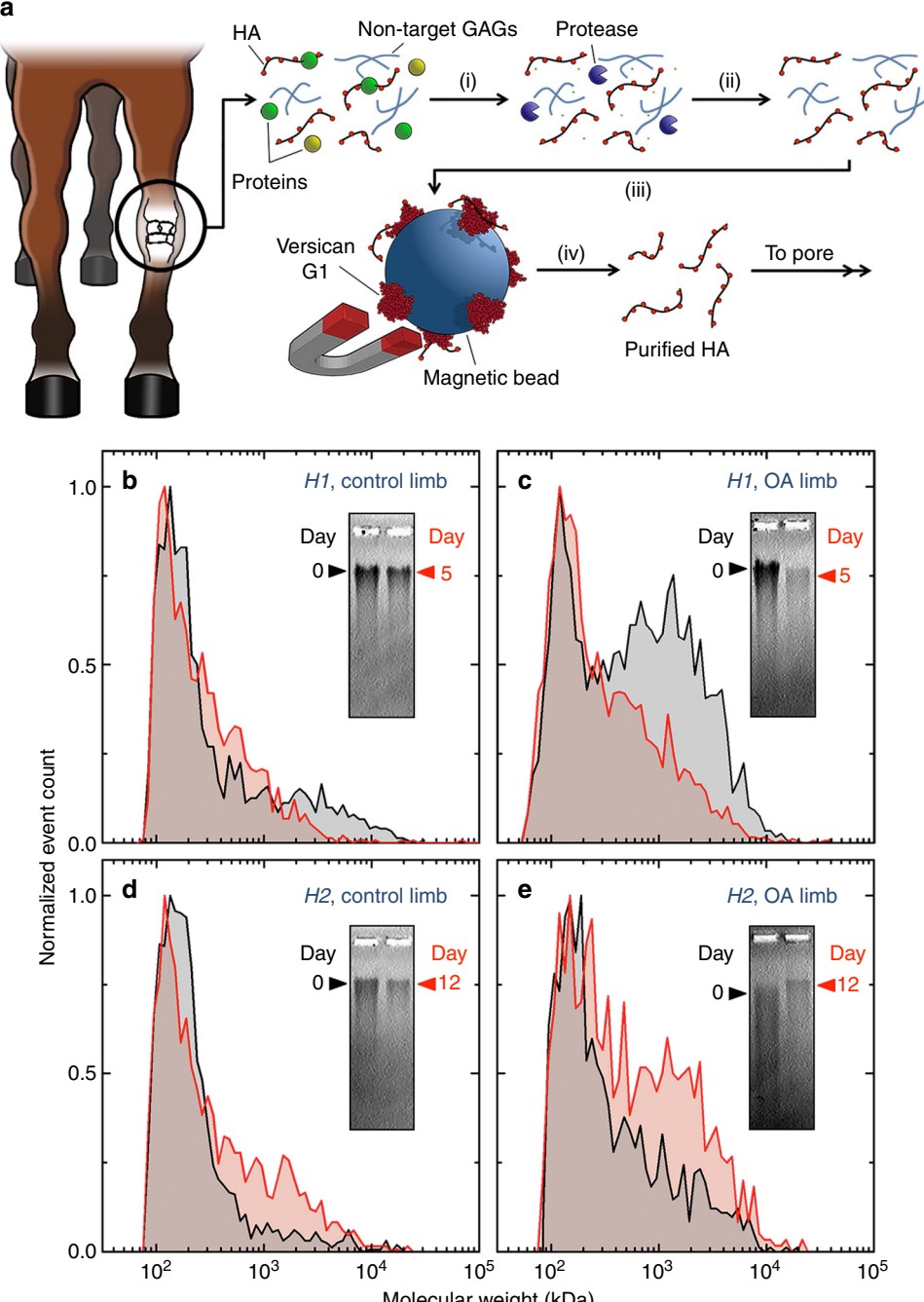

**Fig. 4** Translational analysis of HA derived from equine synovial fluid. **a** Illustration of HA isolation protocol: (i) collected equine synovial fluid is treated with a broad-spectrum protease to digest proteins; (ii) liquid–liquid phase extraction is used to remove protease and remnant protein components; (iii) HA is selectively isolated on versican G1 magnetic beads; and (iv) elution yields pure HA for SS-nanopore analysis. **b–e** Equine synovial fluid HA size distributions obtained from SS-nanopore ECD analysis. For each, day 0 is black and day 5 or 12 is red. Horse *H1* control (sham knee): day 0 ($n = 1768$), day 5 ($n = 1680$); *H1*-induced OA knee: day 0 ($n = 2590$), day 5 ($n = 2849$); *H2* control (sham knee): day 0 ($n = 1748$), day 12 ($n = 1692$); *H2*-induced OA knee: day 0 ($n = 1141$), day 12 ($n = 1215$). Measurements were performed on four different nanopores with diameters ranging from 7.6 to 9.6 nm. Insets show accompanying gel images for the same synovial fluid samples with band positions marked

voltages (Fig. 3d and Supplementary Figure 5), well described over nearly the entire investigated range by a power law fit (Supplementary Table 1) yielding an average exponent $\alpha$ of 2.33 ± 0.16. Only the smallest sample (54 kDa) deviated significantly from this relationship, possibly reflecting the time resolution limits of our current electronics. We expect that HA size differentiation at low-MW ranges could be improved, for example, through the use of high-bandwidth measurement

techniques[34]. The observed power law trend was similar to length dependences measured for other biopolymer translocation durations through SS-nanopores, and was again indicative of the impact of diffusion-limited kinetics. We note that the exponent recovered from our fits (2.33) was higher than previous reports for double-strand DNA[27,31], which ranged from 1.05 to 1.27. This difference could be a result of increased diffusion facilitated by the more compact entropic conformation of HA and reduced

self-avoidance in our high-ionic strength conditions[35]. Regardless, establishment of this trend provided a critical conversion, enabling determination of HA MW at the single-molecule level from the direct electrical output of the SS-nanopore system; for example, using the established relationship as a standard curve, we could determine that the polydisperse HA sample (Fig. 1c) had a mean MW of 311 ± 5 kDa with an interquartile range from about 200–654 kDa.

**Assessment of HA extracted from synovial fluid**. We next applied our approach to the analysis of HA in physiological fluids. Here we focused on synovial fluid, where HA is the major viscoelastic component supporting joint lubrication and hydration[3] and its degradation has been implicated in joint disease. For example, a reduction in HA size and concentration has been associated with OA[36], a common joint pathology that leads to cartilage deterioration. This trend is critical because the viscoelastic and immunomodulatory functions of HA are size-dependent[7]: an observation that positions HA MW distribution in particular as a potentially valuable bioindicator of OA initiation, progression, and treatment efficacy[37]. However, because of the non-selective nature of SS-nanopore signals (i.e., any translocating macromolecule can produce an event), it was not possible to probe synovial fluid without processing to remove other spurious components of biological origin[38]. Therefore, we implemented a procedure (Fig. 4a) for HA isolation that took advantage of the high-binding specificity of the versican protein G1 domain for HA[39].

In our procedure, we first used a broad-spectrum protease to digest protein components of physiological fluid (Fig. 4a, i), including lubricin, collagenases, and especially endogenous HA-binding proteins[40] that could otherwise be retained in the collection scheme. Next, we removed remaining protein (including the exogenously added protease) and lipid components by liquid–liquid phase extraction (Fig. 4a, ii), leaving in solution HA and other aqueous components such as sulfated GAGs[41]. We then incubated the processed mixture with the versican G1 domain immobilized on superparamagnetic beads, followed by magnetic isolation and washing of excess material (Fig. 4a, iii). Finally, bound HA was eluted from the beads thermally (Fig. 4a, iv) to yield a sample suitable for subsequent SS-nanopore analysis. The full protocol typically produced ~150 ng of high-purity HA from 50 μl of raw synovial fluid (Supplementary Table 2).

To test the feasibility and diagnostic potential of the SS-nanopore system, we applied this HA isolation protocol to synovial fluid biospecimens from an established equine model of post-traumatic OA[42] (see Methods section for details). For our initial demonstration of translational SS-nanopore analysis, we focused on two horses (Supplementary Figure 6). For the first (*H1*), conventional gel analysis (Fig. 4b, c, insets) showed a shift in the HA population toward lower MW 5 days after surgical carpal chip induction of OA. This shift is generally indicative of HA degradation, accumulation of low-MW HA fragments, and disease progression, all of which are commonly observed in post-traumatic OA[37,43]. Size distributions obtained by direct conversion of SS-nanopore ECD measurements to MW for the same samples also showed a notable shift in the same direction (Fig. 4b, c), with greater resolution at the lower MW range (<500 kDa) as compared to gels.

A second subject (*H2*) demonstrated an opposite shift toward larger MW after post-traumatic OA induction, as determined by gel analysis (Fig. 4d, e, insets). While OA is known to typically reduce mean HA size through mechanisms that may include joint friction shearing, enzymatic regulation, and immunological

degradation, this effect could in principle be overshadowed by an upregulation of HA synthesis pathways during the acute post-traumatic phase[44] or be affected by natural HA turnover to produce a net increase in MW. This sample provided an experimental counterpoint for our SS-nanopore validation. Indeed, from SS-nanopore size distribution analysis of *H2* (Fig. 4d, e), we observed a notable shift toward higher MW HA 12 days after surgical induction of post-traumatic OA, verifying the results from gel electrophoresis.

We found that the size distributions obtained for sham knees (arthroscopically examined contralateral knees in which no carpal chip was created) were not significantly different from Day 0 to Day 5 or 12, or compared to each other (Fig. 4b, d). This data illustrate the consistency of the measurement across samples and devices. We note that due to the size resolution of the initial SS-nanopore analysis (c.f. Fig. 3d), it is possible that our distribution results overestimate the lowest MW HA in the detectable range and may miss extremely low-MW molecules entirely. This limitation can be improved in future iterations of the system. However, the collective data from the two equine synovial fluid samples presented here are compelling demonstrations of the efficacy of the approach for translational size analysis of HA from biological specimens.

## Discussion

We have presented a SS-nanopore approach for the assessment of the glycan HA, an emerging biomarker with relevance to a broad range of diseases[45]. Through analysis of translocation properties, HA MW can be determined on a per molecule basis, eventually yielding overall size distribution from only a few hundred or thousand individual events. After showing that the platform could detect HA and demonstrating a general capacity to distinguish broad changes in its size distribution, we measured a consistent dependence of event ECD on HA MW using controlled quasi-monodisperse samples. Finally, we developed a general upstream isolation protocol for the specific isolation of HA from biological fluids toward the purpose of SS-nanopore HA size distribution determination in synovial fluid from an equine post-traumatic OA model. Such a sample in our prototype device consisted of as little as 10 ng of HA in a 10 μl volume, which could be measured electrically in ~2 h. This time could be shortened significantly by using higher HA concentrations.

This study establishes SS-nanopores as a tool for the analysis of HA, demonstrating high quality, reliable, and reproducible (Supplementary Figures 7-9) quantitative data on both HA detection and size distribution determination from biological specimens. The sensitivity, speed, and small sample volume requirements of this approach make it attractive as the basis for future diagnostic tools with distinct advantages over conventional glycan assessment technologies. Applications for the technology may include both translational measurement of HA as a biomarker, as well as assessment of HA synthesis products for commercial or research purposes. The results also suggest a wider role for the measurement platform in assessing other important glycans, GAGs, and proteoglycans that may have additional importance as bioindicators of diverse pathologies, including heparan sulfate[46], chondroitin sulfate[47], and keratan sulfate[48].

## Methods

**HA samples**. Purified polydisperse *Streptococcus zooepidemicus* HA (Vesta, Indianapolis, IN) was mixed as received in deionized water to a concentration of 1 μg/μl as a bulk solution; no further purification was performed. Discrete quasi-monodisperse HA samples[33] were provided by Hyalose, LLC. (Oklahoma City, OK). A total of seven quasi-monodisperse HA samples (54, 81, 130, 237, 545, 1076, and 2384 kDa) were used, with MW within 5% of the reported mean (polydispersity = 1.001–1.035, as estimated by MALLS-SEC). Each 50 μg lyophilized sample was mixed with deionized water to produce a 1 μg/μl solution. All samples

were stored in LoBind Eppendorf tubes (Fisher Scientific, Hampton, NH) at 4 °C for short term use, or kept at −20 °C for long-term storage.

**Ultrasonic shearing of polydisperse HA**. A 50 μl solution of polydisperse HA concentrate (1 μg/μl) was placed in a microTUBE AFA fiber snap-cap (Covaris, Woburn, MA) and mechanically sheared in a 7 °C water bath using a Covaris S220 focused ultrasonicator (peak incident power of 175 W, 200 cycles per burst, 10% duty factor). Shearing was varied by increasing sonication times in 5 s increments. HA fragmentation was monitored by gel electrophoresis using the methods described below.

**Gel electrophoresis of HA**. Electrophoresis was conducted on a 0.5% agarose gel in 1× TAE buffer. All samples (polydisperse and quasi-monodisperse HA) were aliquoted as 12 μl volumes in 0.15 NaCl solution using a minimum of 1–3 μg HA for visualization, consistent with previous literature[49,50]. For synovial fluid samples, collected material was centrifuged at 300×g for 5 min at 4 °C to pellet the cellular material, and the supernatant was retrieved and stored at −80 °C. Prior to gel electrophoresis, the solution was thawed, diluted 1:20 in PBS buffer, and incubated with proteinase K (1 mg/mL) overnight to digest protein components. The resulting mixture was loaded directly onto gel because analyte visualization was insensitive to the trace background components. Electrophoresis was performed at 34 V for 3.5 h at room temperature for polydisperse and quasi-monodisperse HA samples, and at 50 V for 8 h at room temperature for synovial fluid HA samples. Staining was performed as described previously[49]. Briefly, the gel was submerged overnight in a room temperature bath of 0.005% Stains-All (Sigma-Aldrich, St. Louis, MO) in 50% ethanol, which was taken to prevent light exposure. Next, destaining was performed by incubating the gel in 10% ethanol for 8 h, still in the dark. The bath was refreshed with clean solution at least twice during this time. Finally, the gel was removed and excess ethanol solution was removed manually by wicking with laboratory wipes. Gel images were collected under white light tran-sillumination using a ChemiDoc XRS + system (BioRad, Hercules, CA) for Figs. 2–3 and a VersaDoc system (BioRad) for Fig. 4. Migration distance was determined via image analysis (ImageJ)[51] by determining the distance from the bottom of the loading well for each band.

**SS-nanopore preparation**. Silicon chips (4 mm) with a thin, free-standing silicon nitride (SiN) membrane (8–2 μm with 25 nm thickness) were obtained commer-cially (Norcada, Inc. Alberta, Canada) for solid-state nanopore fabrication. Indi-vidual pores were formed in-house using an Orion Plus helium ion microscope (Carl Zeiss, Peabody, MA) following methods described elsewhere[52]. Briefly, the focus and astigmatism of a focused helium beam were first optimized on an area of the silicon chip near the suspended SiN window using point exposures. Then, the beam position was blanked, moved to the center of the SiN membrane, and exposed for a calibrated time to produce a single pore with reproducible dimen-sions. All nanopores used in this work were formed with diameters in the range of 6.5–8.6 nm. Following fabrication, chips were stored in 50% ethanol solution prior to use. In preparation for measurement, each nanopore chip was rinsed with DI water and absolute ethanol, then dried with filtered air, and subsequently exposed to a 30 W air plasma (Harrick Plasma, Ithaca, NY) for two minutes on each side before being positioned in a custom Ultem 1000 flow cell. Measurement buffer (6 M LiCl, 10 mM Tris, 1 mM EDTA, pH 8.0) was then introduced to both sides of the chip. Prior to use, prepared buffers were treated in an ultrasonic bath (FS20, Fisher Scientific) for 5 min and then passed through a 0.45 μm syringe filter (Minisart NY25, Sartorius, Bohemia, NY) to remove contaminants and pre-cipitates. Ag/AgCl electrodes (Sigma-Aldrich, St. Louis, MO) were positioned in each chamber for voltage application and ionic current measurement using an Axopatch 200B patch clamp amplifier (Molecular Devices, Sunnyvale, CA). Each chip was pre-checked using clean measurement buffer to ensure a steady, low-noise baseline current with no spurious events and a linear current–voltage (I–V) curve that verified SS-nanopore diameter in assessment buffer (1 M NaCl, 10 mM Tris, 1 mM EDTA) using an established model[31] modified to incorporate empirical conductivity of high-concentration electrolytes in aqueous solution[53]. Pore dia-meters were stable, varying by <1 nm over typical measurement times (Supple-mentary Figure 10).

**SS-nanopore analysis of HA**. Prior to HA analysis, the assessment buffer was exchanged for measurement buffer to maximize signal-to-noise ratio[54]. HA was loaded by pipetting 10–20 μl into one flow cell chamber at a final concentration of 50 ng/μl unless otherwise noted. The data were collected at a rate of 200 kHz with a four-pole Bessel filter designed to be 100 kHz, but actually corresponding[55] to 57 kHz. Analysis was performed using custom software, with which an additional 5 kHz low-pass filter was applied to all collected data. Each sample was tested in a series of trials at voltages ranging typically from 100–400 mV. Event threshold was defined as a deviation of at least five standard deviations (5σ) from baseline current with a duration between 25 μs and 2.5 ms. HA translocation was confirmed via recapture events[24] (Supplementary Figure 1), as well as observation of voltage-dependent reductions in event duration for quasi-monodisperse HA (Supple-mentary Figure 11). ECD was calculated for each deviation as the area

encompassing the event[28] by integrating the nanopore current for the duration of time it remained beyond the 5σ threshold value. Event rates were determined from uninterrupted current traces of 3.2 s increments at a single condition. The standard deviation measured between increments was used as an indication of measurement error.

**Equine model of osteoarthritis**. Equine synovial fluid was obtained from healthy adult horses (2–5 years old) with radiographically normal carpal joints. Post-traumatic osteoarthritis was induced surgically through a carpal chip defect in one randomly assigned forelimb[42]. Briefly, an 8 mm osteochondral fragment was cre-ated in the dorsal rim of the radial carpal bone and left within the joint. The exposed subchondral bone was then debrided using an arthroburr to generate a 15 mm defect; the debris generated from the procedure was not removed from the synovial cavity. A sham operation (arthroscopic examination without carpal chip induction) was performed on the contralateral leg to serve as a control. Two weeks after the induced osteochondral fragmentation, without operative intervention, the horses were subjected to a 30 min treadmill/5 days per week training regimen to initiate post-traumatic osteoarthritis (OA), and synovial fluid samples were col-lected from both joints on Day 0 as well as Day 5 (H1) or 12 (H2) post-surgery. Samples were kept at −80 °C prior to use. All animal and tissue collecting protocols were approved by the Cornell University's Institutional Animal Care and Use Committee (Protocol Number: 2012-0097).

**High-purity HA extraction from synovial fluid**. HA was isolated from the equine synovial fluid using a protocol adapted from Yuan et al.[50]. Raw equine synovial fluid (50 μl) was first incubated with 1.8 U/mL proteinase K (New England Biolabs, Ipswich, MA) for 15 min at 37 °C to digest protein components, including those with HA-binding capacity. An equal volume of a phenol:chloroform:isoamyl alcohol (25:24:1 v/v/v, Fisher Scientific) was then added to the sample and mixed thoroughly before being centrifuged for 15 min at 14,000×g in a Phase Lock Gel Tube (QuantaBio, Beverly, MA) to separate the aqueous HA from the organic component. This extraction process was repeated once using pure chloroform to remove residual phenol from the aqueous phase, which was found to adversely affect downstream protein function during the affinity-based purification steps.

In preparation for the isolation of pure HA, first streptavidin magnetic beads (Dynabeads M-280, Invitrogen, Carlsbad, CA) at a concentration 10 mg/mL were washed three times with 1× PBS, 0.05% Tween by adding buffer, mixing gently, and aspirating under magnetic field, and then three times with 1× PBS only. After washing, 250 μl of packed beads were resuspended in 50 μl of 1× PBS. Then, 21 μl of biotinylated versican G1 domain (bVG1, 1.23 μg/μl, Echelon Biosciences, Salt Lake City, UT) was added directly to the beads, mixed, and incubated for 1 h at room temperature on a rocker. After incubation, the beads were washed three times with 150 μl 1× PBS to remove any unbound bVG1.

The bVG1-streptavidin beads were subsequently reconstituted with the solvent-extracted HA solution and incubated at room temperature for 24 h with gentle rocking. The sample was placed on a magnet to pull down the beads (with bound HA) and the supernatant was aspirated. The beads were washed three times with 1× PBS, after which deionized water was added to the sample to a final volume of 50 μl. To denature the bVG1 and release the bound HA, the sample was placed on a heating block at 95 °C for 15 min. Finally, the vial was placed on a magnet and the solution containing released, purified HA was removed and stored at −20 °C until use. Yield was determined through direct quantification of the isolate with an enzyme-linked immunosorbent assay kit (HA ELISA, Echelon Biosciences; Supplementary Table 2).

**Data Availability**. The authors declare that the data supporting the findings of this study are available within the paper and its Supplementary Information files.

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

## Acknowledgements

The authors wish to acknowledge Courtney Smith for early contributions to the measurements and Glenn Prestwich for helpful discussions. This work was supported by start-up funds to E.R. and A.R.H. from Wake Forest University Health Sciences.

## Author contributions

F.R. performed biochemical treatments, handled physiological fluid samples, performed SS-nanopore measurements, and analyzed data. O.K.Z. contributed to experimental design and performed initial SS-nanopore measurements. H.L.R., B.T.P., and A.J.N. collected synovial fluid samples and performed gel analyses. P.L.D., E.R., and A.S. provided materials, contributed to experimental design, and edited the manuscript. E.R. and A.R.H. oversaw the project, contributed to data analysis, and wrote the manuscript. All authors reviewed the manuscript.

## Additional information

**Competing interests:** A.R.H, E.R., and P.L.D. are listed as inventors on a provisional patent involving the described technology. Remaining authors declare no competing interest.

