## [Peer Review File · Nature Communications]

Reviewers' Comments:

Reviewer #1:

Remarks to the Author:

This paper from Hall and coworkers is an interesting study in which the authors use solid-state nanopores to detect small quantities of label-free hyaluronic acid macromolecules both from synthetic samples to assess the feasibility of the method and from synovial fluid samples. The research presented in this study is clearly novel as it is the first study, as far as I know, that uses solid-state nanopores to detect and analyze the size distribution of large molecular weight HA polysaccharides. The authors showed that the presence of as little as 10 ng of HA in synovial fluids can be detected using this measurement platform and that it could also be used to detect other important glycosaminoglycans, related to diverse pathologies. This study is of broad interest, especially considering the fact that polysaccharide molecules have very complex structures and all the different analytical methods currently used have significant limitations as pointed out by the authors. The manuscript is well written and is overall clear to follow and to understand. The manuscript gives a complete description of the procedures that could be reproduced by others in the field.

I would be in favor of publication after minor revisions:

p3: The authors say that only small HA (<10 individual sugar residues) could be measured with an aerolysin nanopore. The 10 individual sugar residues limit is the upper limit that Fennouri et al. showed that they could discriminate but they also showed that an enzymatic reaction resulted in much longer events (especially after short reaction times). These events are probably related to polysaccharides longer than 10 residues. The detection limit for protein nanopores is however most likely much lower than the limit that could be reached by the authors in this study using solid-state nanopores. This should be modified in the main text.

p4: The authors state that they could measure recapture HA events after their translocation through a nanopore with data showing evidence for this in the supplement. This has already been done in the past by Gershow & Golovchenko, Nature Nanotechnology 2, 775 - 779 (2007) with DNA strands and this reference should be mentioned.

p6: The authors say that "only the smallest sample (54 kDa) deviated significantly from this relationship". In the caption of Supplementary Figure S2, they say that "the solid lines are power-law fits to data excluding the lowest MW (54 kDa)" which is consistent with what is written in the main text but in the caption of Figure 3 they say that the "solid line is a power-law fit ($\alpha=2.15$) to the data down to 130 kDa". Why is the point at 81 kDa not included in this power-law fit?

p6: The authors say that by using the established relationship as a standard curve, they could estimate the mean MW of the polydisperse HA sample to be around 250 kDa. This is clear when looking at the standard curve but can the authors provide the equation of that curve? According to Figure 3d, and using the only information provided in the caption ($\alpha=2.15$), the equation of the curve would be in the form of $ECD = [MW]^{2.15}$. If I use the approximated value of 250 kDa for the mean MW of the polydisperse HA sample, the resulting ECD would be approximately $4E11$ ke. This equation seems to work if the MW is entered in kDa instead of Da like stated in the x axis of the graph. Either the x axis of the graph should be corrected or the units for parameters should be mentioned.

p7: The authors show that 5 days after surgical carpal chip induction of OA to two horses, HA population showed a shift toward lower or higher MW. It would be nice to include typical traces (maybe in the supplement) for the different conditions.

p8: The authors say that "While OA is known to typically reduce mean HA size through joint friction shearing, this effect can in principle be overshadowed by an upregulation of HA synthesis

pathways during the acute post-traumatic phase or be affected by natural HA turnover to produce a net increase in MW." Could the authors give references for these two statements?

p8: The difference between the control limbs (the sham knee of the horses) and the OA limbs is slightly confusing after the first read, especially to a reader unfamiliar with animal procedures. It would be nice to explain clearly in the main text how the sham knees are different maybe only with a few words in parentheses.

p11: In the Materials & Methods section, the authors say that they use a 100 kHz four-pole Bessel filter with an Axopatch 200B patch clamp amplifier. A previous study (Uram et al., Noise and Bandwidth of Current Recordings from Submicrometer Pores and Nanopores, ACS Nano, 2008, 2 (5), pp 857–872) showed that even though the manufacturer of the amplifier specifies that the maximum cutoff frequency is 100 kHz, this in reality corresponds to a signal bandwidth of 57 kHz. The manufacturer states that this signal bandwidth is lower than expected for a low-pass Bessel filter with a cutoff frequency of 100 kHz due to the signal bandwidth limitation of the electronics in the headstage of the amplifier. The authors should consider adding this for more transparency.

p21: In Figure 2b, the authors show a histogram of the ECD values and say that the solid lines correspond to Gaussian fits to the data. The x axis is however in a logarithmic scale. A LogNormal fitting to the x values seems to be more appropriate here. Can the authors explain more in details how they fitted the data?

p22: In Figure 3c and d, the applied voltage should be mentioned since three different applied voltages are shown in the supplement.

Supplementary Figure S2: The fit to the data recorded at 400 mV seems incorrect if the lowest MW point is excluded. Can the authors check that the fitting has been performed correctly? The fit functions and how the fits have been done should also be included for all conditions.

Reviewer #2:

Remarks to the Author:

- What are the major claims of the paper?

The paper demonstrates hyaluronic acid's (HA) detection and weight discrimination, which ranges from 54 kDa to 2.4 MDa. Using mono and poly-disperse HA the authors show a correlation between event charge deficit (ECD, a method previously published in 2005 by Fologea et al) from current drop events using solid-state nanopores, and established analytical methods (gel electrophoresis) to discriminate size distributions and average molecular weights. Moreover, by following size distributions over time, they use HA purified from synovial fluid as a biomarker of equine osteoarthritis. The paper is well written and interesting but needs improvement especially on the reproducibility of the data and nanopore conditions (no details provided).

- Are the claims novel? If not, please identify the major papers that compromise novelty
Yes. The authors are able to detect HA as well as discriminate HA size using a solid-state nanopore. The only novelty here is the use of solid-state nanopores compared to protein nanopores. Those experiments are complicated.

- Will the paper be of interest to others in the field?

Yes

- Will the paper influence thinking in the field?

No, it only shows that, using solid-state nanopores, a concept already applied for DNA can be applied to other polymers.

- Are the claims convincing? If not, what further evidence is needed?
 - As a proof of real translocation, one would expect to see the dependencies of dwell times as a function of the applied voltage.
 - The authors should provide details of current events showing folded and unfolded molecules in their supplementary information section. It would also be beneficial for a general audience for them to explain why they use ECD.

- Are there other experiments that would strengthen the paper further? How much would they improve it, and how difficult are they likely to be?
No

- Are the claims appropriately discussed in the context of previous literature?
Yes

- Is the manuscript clearly written? If not, how could it be made more accessible?
The manuscript is clearly written and easy to read.

- Could the manuscript be shortened to aid communication of the most important findings?
No, this is well written.

- Have the authors done themselves justice without overselling their claims?
No. In the conclusion: "HA MW can be determined on a per molecule basis, eventually yielding overall size distribution from only a few hundred or thousand individual molecules". Do the authors mean "few hundred or thousand individual events"? This is clearly different. The authors need to discuss stability of the pore past 2 hours of experiment. Is the pore expending? Is the pore stable? Please provide more details (current traces over time) on pore stability.
The conclusion claims: "reproducible quantitative data on both HA detection and size distribution...". As mentioned below, the authors need to provide more details on reproducibility of the experiments.

- Have they been fair in their treatment of previous literature?
Yes

- Have they provided sufficient methodological detail that the experiments could be reproduced?
 - How many independent experiments have been performed for each condition? (detection of polydisperse HA, assessment of HA mechanical shearing, analysis of monodisperse HA and translational analysis of HA from synovial liquid). It is necessary to show that the data are reproducible depending on the pore used (geometry, size etc...). It seems from the manuscript that only one pore has been used per condition.
 - Furthermore, for each independent experiment, the authors need to provide nanopore size and membrane thickness to be able to accurately reproduce the data. This can be easily fixed in the figure legends.
 - A single line could also be added in the figure legend on the salt conditions used.

- Is the statistical analysis of the data sound?
Yes, if the authors provide evidence of data reproducibility with different pores for each condition. The authors need to provide more information on the fits (for example in the figure legend). In figure 2 b, data were fit by gaussians. The authors need to give the fit values. In supplementary Figure S2, please give error bars for power law fits. Same for the table S1. Page 6, the mean MW of the polydisperse HA should also have error bars. Please correct throughout the manuscript.

- Should the authors be asked to provide further data or methodological information to help others

replicate their work? (Such data might include source code for modelling studies, detailed protocols or mathematical derivations).

Yes, in supplementary information or methods section the authors should provide more details on how they obtained ECD data. For a journal like Nature Communications, the authors should consider the potential breadth of their audience and assume they are not necessarily acquainted with nanopore technology.

There is no information on the net charge or pI of HA. It would be useful when proving that translocation is governed by electrophoresis. It is only written "negatively charged molecules".

- Are there any special ethical concerns arising from the use of animals or human subjects?
No

Reviewer #3:

Remarks to the Author:

This work describes the use of solid-state nanopores as sensors to detect hyaluronan. I enjoyed reading this manuscript as I think probing of non-DNA medically relevant biopolymers is an area where SS-nanopores could excel over other approaches especially at higher MWs. The work was thorough and the manuscript was well written. I don't think any major revisions are required and I recommend the work be published with some minor additions. I have some minor questions/comments listed below. I don't think the authors need to do additional experiments for this paper, some of the following are just things the authors should think about.

1) It would be nice to see dwell time histograms for the quasi-monodisperse HA samples. Do these have a long tail? The authors mention that clogging was only a major issue at high concentrations and high voltage. Do the authors see long duration events for the low MW samples, indicative of sticking?

2) At the higher event rates, for the longer dwell time events, what is the Poisson probability of two molecules translocating at the same time? In a polydisperse sample co-translocation of a high MW HA with a low MW HA would look just like a slightly higher MW HA (with a fold).

3) How reproducible are the ECD distributions from pore to pore? If I measure a standard curve on one pore, how well would it work on another? The authors suggested that once measured the standard curve could be used for subsequent measurements, but it would be nice to have a bit more to back this up or clarify their statements.

4) Most people in the nanopore field are used to thinking about DNA as it is the most common polymer so adding some of the polymer properties of hyaluronan would be nice (if known), like linear charge density, contour length or end-to-end distance vs MW, persistence length,... The amount of folding and translocation dynamics will depend on these properties. A quick search gives a persistence length of ~5nm coupled with the very high ionic strength, should I be thinking of HA as a highly compact blob in the LiCl buffer? It would be nice to see a couple of events zoomed in, perhaps for the high MW HA. Do these look like dsDNA events with folding (given the quantized current histograms, as the authors point out) or are they more like ssDNA events? Does the HA look like it's captured on end or along the molecule?

5) The author's briefly allude at bandwidth issues. How is their SNR at 10kHz?

Not for this manuscript, but as they do future work I would urge the authors to consider:

i) Does the bead immunoprecipitation introduce selection bias in the polymer length distribution? There are several ways to check this, one could be mixing quasi-monodisperse HA samples and measuring them before and after bead immunoprecipitation.

ii) In my experience with LiCl buffers >4M LiCl, we would often see spurious translocation events in the blank buffer control, which we attributed to nano-size salt precipitate, even though the solubility of LiCl is in the high 15M range. These would be visible under DLS measurements. I would urge the authors to be on the lookout for these when using very high molarity LiCl buffers.

Reviewers' comments

(author responses in blue)

Reviewer #1 (Remarks to the Author):

This paper from Hall and coworkers is an interesting study in which the authors use solid-state nanopores to detect small quantities of label-free hyaluronic acid macromolecules both from synthetic samples to assess the feasibility of the method and from synovial fluid samples. The research presented in this study is clearly novel as it is the first study, as far as I know, that uses solid-state nanopores to detect and analyze the size distribution of large molecular weight HA polysaccharides. The authors showed that the presence of as little as 10 ng of HA in synovial fluids can be detected using this measurement platform and that it could also be used to detect other important glycosaminoglycans, related to diverse pathologies. This study is of broad interest, especially considering the fact that polysaccharide molecules have very complex structures and all the different analytical methods currently used have significant limitations as pointed out by the authors. The manuscript is well written and is overall clear to follow and to understand. The manuscript gives a complete description of the procedures that could be reproduced by others in the field.

I would be in favor of publication after minor revisions:

p3: The authors say that only small HA (<10 individual sugar residues) could be measured with an aerolysin nanopore. The 10 individual sugar residues limit is the upper limit that Fennouri et al. showed that they could discriminate but they also showed that an enzymatic reaction resulted in much longer events (especially after short reaction times). These events are probably related to polysaccharides longer than 10 residues. The detection limit for protein nanopores is however most likely much lower than the limit that could be reached by the authors in this study using solid-state nanopores. This should be modified in the main text.

We have amended the text as follows: “However, the dynamic range of that system is very narrow; direct assessment was limited to small HA (<10 individual sugar residues) with only indirect evidence of larger molecule detection.”

p4: The authors state that they could measure recapture HA events after their translocation through a nanopore with data showing evidence for this in the supplement. This has already been done in the past by Gershow & Golovchenko, Nature Nanotechnology 2, 775 - 779 (2007) with DNA strands and this reference should be mentioned.

Reference has been added

p6: The authors say that “only the smallest sample (54 kDa) deviated significantly from this relationship”. In the caption of Supplementary Figure S2, they say that “the solid lines are power-law fits to data excluding the lowest MW (54 kDa)” which is consistent with what is written in the main text but in the caption of Figure 3 they say that the “solid line is a power-law fit ($\alpha=2.15$) to the data down to 130 kDa”. Why is the point at 81 kDa not included in this power-law fit?

We have re-checked our data and fits, and confirmed that this was a typo: the fit excludes only the lowest MW (54 kDa), as described in the main text. We have revised the caption to reflect this. We have also redone the fits for consistency, resulting in minor changes to the listed α (see Table S1 and below for details).

p6: The authors say that by using the established relationship as a standard curve, they could estimate the mean MW of the polydisperse HA sample to be around 250 kDa. This is clear when looking at the standard curve but can the authors provide the equation of that curve? According to Figure 3d, and using the only information provided in the caption ($\alpha=2.15$), the equation of the curve would be in the form of $ECD=$

[[MW]] ^2.15. If I use the approximated value of 250 kDa for the mean MW of the polydisperse HA sample, the resulting ECD would be approximately 4E11 ke. This equation seems to work if the MW is entered in kDa instead of Da like stated in the x axis of the graph. Either the x axis of the graph should be corrected or the units for parameters should be mentioned.

We have changed the x-axis for Fig. 3d to be in 'kDa' and supplied the fit equation and parameters in Table S1. We thank the reviewer for this detailed inspection.

p7: The authors show that 5 days after surgical carpal chip induction of OA to two horses, HA population showed a shift toward lower or higher MW. It would be nice to include typical traces (maybe in the supplement) for the different conditions.

Concatenated translocation events for each of the conditions have now been added as a supplemental figure (Fig. S6).

p8: The authors say that "While OA is known to typically reduce mean HA size through joint friction shearing, this effect can in principle be overshadowed by an upregulation of HA synthesis pathways during the acute post-traumatic phase or be affected by natural HA turnover to produce a net increase in MW." Could the authors give references for these two statements?

We have added a reference to Chan, et al. (doi: 10.1016/j.joca.2015.06.021), which reports on upregulation of hyaluronan synthase in a murine model of knee joint cartilage damage

p8: The difference between the control limbs (the sham knee of the horses) and the OA limbs is slightly confusing after the first read, especially to a reader unfamiliar with animal procedures. It would be nice to explain clearly in the main text how the sham knees are different maybe only with a few words in parentheses.

We have added the following statement to our first mention of the sham knee (pg. 8) for clarification: "...sham knees (arthroscopically examined contralateral knees in which no carpal chip was created)..."

p11: In the Materials & Methods section, the authors say that they use a 100 kHz four-pole Bessel filter with an Axopatch 200B patch clamp amplifier. A previous study (Uram et al., Noise and Bandwidth of Current Recordings from Submicrometer Pores and Nanopores, ACS Nano, 2008, 2 (5), pp 857–872) showed that even though the manufacturer of the amplifier specifies that the maximum cutoff frequency is 100 kHz, this in reality corresponds to a signal bandwidth of 57 kHz. The manufacturer states that this signal bandwidth is lower than expected for a low-pass Bessel filter with a cutoff frequency of 100 kHz due to the signal bandwidth limitation of the electronics in the headstage of the amplifier. The authors should consider adding this for more transparency.

We have revised the Materials & Methods section on *SS-nanopore analysis of HA* (pg. 11) to read, "Data was collected at a rate of 200 kHz with a four-pole Bessel filter designed to be 100 kHz, but actually corresponding to 57 kHz", and added the suggested citation. Because subsequent digital filtering is performed, this does not affect our results.

p21: In Figure 2b, the authors show a histogram of the ECD values and say that the solid lines correspond to Gaussian fits to the data. The x axis is however in a logarithmic scale. A LogNormal fitting to the x values seems to be more appropriate here. Can the authors explain more in details how they fitted the data?

We thank the reviewer for catching this oversight; this was a Gaussian fit on a semi-log scale, and so an accurate description would indeed be 'log-normal'. We have revised the manuscript accordingly.

p22: In Figure 3c and d, the applied voltage should be mentioned since three different applied voltages are shown in the supplement.

Applied voltage has been added to the caption.

Supplementary Figure S2: The fit to the data recorded at 400 mV seems incorrect if the lowest MW point is excluded. Can the authors check that the fitting has been performed correctly? The fit functions and how the fits have been done should also be included for all conditions.

We have reviewed all of the power law fits presented in both the manuscript and supplement, and confirmed that all were performed across the same range of MW (81 kDa and up). However, we did find that slightly different fitting methods were used (least squares vs. damped least squares (DLS)). For consistency, we have refit all data using DLS fitting. This change produced only minor differences in the fitting results. However, this did require us to also re-plot the equine SF size distribution plots in Fig. 4, which used the fit as a direct transformation; this re-plot also did not change results meaningfully.

Reviewer #2 (Remarks to the Author):

• What are the major claims of the paper?

The paper demonstrates hyaluronic acid's (HA) detection and weight discrimination, which ranges from 54 kDa to 2.4 MDa. Using mono and poly-disperse HA the authors show a correlation between event charge deficit (ECD, a method previously published in 2005 by Fologea et al) from current drop events using solid-state nanopores, and established analytical methods (gel electrophoresis) to discriminate size distributions and average molecular weights. Moreover, by following size distributions over time, they use HA purified from synovial fluid as a biomarker of equine osteoarthritis. The paper is well written and interesting but needs improvement especially on the reproducibility of the data and nanopore conditions (no details provided).

• Are the claims novel? If not, please identify the major papers that compromise novelty

Yes. The authors are able to detect HA as well as discriminate HA size using a solid-state nanopore. The only novelty here is the use of solid-state nanopores compared to protein nanopores. Those experiments are complicated.

• Will the paper be of interest to others in the field?

Yes

• Will the paper influence thinking in the field?

No, it only shows that, using solid-state nanopores, a concept already applied for DNA can be applied to other polymers.

We would like to bring attention to the reviewer that while the fundamental principle of single molecule sensing with solid-state nanopores is not new, the use of these pores for glycan sensing can certainly influence and impact the current nanopore and glycosciences fields.

• Are the claims convincing? If not, what further evidence is needed?

- As a proof of real translocation, one would expect to see the dependencies of dwell times as a function of the applied voltage.

We have included analysis of quasi-monodisperse HA samples as a function of voltage as Fig. S11. As the reviewer suggests, we observe reductions in event duration as applied voltage is increased.

- The authors should provide details of current events showing folded and unfolded molecules in their

supplementary information section. It would also be beneficial for a general audience for them to explain why they use ECD.

We have revised the text to make the utility of ECD to this work clearer, first by describing the significance of event amplitude and duration separately (pg. 3) and then by justifying the choice of ECD over event duration explicitly. In addition, we have added a series of typical unfolded and folded event traces as Fig. S2 to highlight the differences in shape that drove the use of ECD, as suggested.

• Are there other experiments that would strengthen the paper further? How much would they improve it, and how difficult are they likely to be?

No

• Are the claims appropriately discussed in the context of previous literature?

Yes

• Is the manuscript clearly written? If not, how could it be made more accessible?

The manuscript is clearly written and easy to read.

• Could the manuscript be shortened to aid communication of the most important findings?

No, this is well written.

• Have the authors done themselves justice without overselling their claims?

No. In the conclusion: "HA MW can be determined on a per molecule basis, eventually yielding overall size distribution from only a few hundred or thousand individual molecules". Do the authors mean "few hundred or thousand individual events"? This is clearly different.

We have clarified this sentence so as not to mislead the reader and thank the reviewer for pointing it out. The revised sentence now reads, "Through analysis of translocation properties, HA MW can be determined on a per molecule basis, eventually yielding overall size distribution from only a few hundred or thousand individual events".

The authors need to discuss stability of the pore past 2 hours of experiment. Is the pore expanding? Is the pore stable? Please provide more details (current traces over time) on pore stability.

In response to this concern, we have performed additional experiments to demonstrate the stability of our pores by keeping SS-nanopore devices under conditions identical to those used during HA measurement (6 M LiCl, 10 mM Tris, 1 mM EDTA, pH 8.0, at an applied voltage of 200 mV) for up to five hours and reassessing the pore diameter every 30 min. Supplementary Fig. S10 shows a typical result for a device with an initial pore diameter of 8.9 nm. Over the long-term experiment, the pore grows less than 1 nm.

In addition, we include here a plot of the power spectral density taken both at the beginning and at the end of a single, long-term HA measurement (Fig. R1), showing no significant difference in the noise.

Figure R1. Noise comparison Power spectral density of the same SS-nanopore at the beginning (blue) and the end (red) of a HA measurement

The conclusion claims: “reproducible quantitative data on both HA detection and size distribution...”. As mentioned below, the authors need to provide more details on reproducibility of the experiments.

We respond to this in detail below.

• Have they been fair in their treatment of previous literature?

Yes

• Have they provided sufficient methodological detail that the experiments could be reproduced?

- How many independent experiments have been performed for each condition? (detection of polydisperse HA, assessment of HA mechanical shearing, analysis of monodisperse HA and translational analysis of HA from synovial liquid). It is necessary to show that the data are reproducible depending on the pore used (geometry, size etc...). It seems from the manuscript that only one pore has been used per condition.

We have performed the measurements multiple times and on multiple pores with a range of diameters. To address this comment directly, we have added additional data including: (i) comparative polydisperse HA analysis (Fig. S7), showing statistically indistinguishable results obtained from two typical SS-nanopores (diameters 6.2 and 7.4 nm, respectively); and (ii) comparative monodisperse HA analyses (for 130 and 237 kDa, Fig. S8), showing statistically indistinguishable results obtained from two SS-nanopores each (diameters 7.6 to 8.6 nm). In addition, we provide a compiled plot of HA MW vs ECD for multiple SS-nanopores (Fig. S9) to demonstrate the consistency of our data.

- Furthermore, for each independent experiment, the authors need to provide nanopore size and membrane thickness to be able to accurately reproduce the data. This can be easily fixed in the figure legends.

Nanopore diameters have been added to all pertinent figures in the paper. All results in the main manuscript used the same membrane thickness (noted in Materials and Methods), and so we did not add this explicitly to

the captions. Both diameter and thickness are noted in all pertinent spots in the supplemental information.

- A single line could also be added in the figure legend on the salt conditions used.

The same buffer conditions (6 M LiCl, 10 mM Tris, 1 mM EDTA, pH 8.0) were used for all measurements presented in this paper. This is indicated under Materials and Methods. If the editors prefer it be added to individual captions as well, we will be happy to do so, but we have not in this revision to avoid repetition.

• Is the statistical analysis of the data sound?

Yes, if the authors provide evidence of data reproducibility with different pores for each condition.

Please see above.

The authors need to provide more information on the fits (for example in the figure legend). In figure 2 b, data were fit by gaussians. The authors need to give the fit values.

We have provided detailed parameter results for all log-normal (Gaussian on a log scale) fits in Table S1.

In supplementary Figure S2, please give error bars for power law fits.

All errors for power law fit parameters are now provided in Table S1.

Same for the table S1.

We have added error to the final (mean HA mass) column and noted in the caption <10% systematic error is associated with ELISA in general.

Page 6, the mean MW of the polydisperse HA should also have error bars. Please correct throughout the manuscript.

We have amended the mean polydisperse HA MW value (pg. 6 of the manuscript) as requested. We do not provide this at any other point in the paper.

• Should the authors be asked to provide further data or methodological information to help others replicate their work? (Such data might include source code for modelling studies, detailed protocols or mathematical derivations).

Yes, in supplementary information or methods section the authors should provide more details on how they obtained ECD data. For a journal like Nature Communications, the authors should consider the potential breadth of their audience and assume they are not necessarily acquainted with nanopore technology.

We have expanded the description in the Materials and Methods section as follows: "Event Charge Deficit (ECD) was calculated for each deviation as the area encompassing the event by integrating the nanopore current for the duration of time it remained beyond the 5 σ threshold value."

There is no information on the net charge or pI of HA. It would be useful when proving that translocation is governed by electrophoresis. It is only written "negatively charged molecules".

We have added both requested values to the revised manuscript. We calculate the linear charge density from

the chemical structure of HA disaccharide units (1 e⁻ charge, 1 nm length) as -0.16 nC/m. For comparison, the value for double-strand DNA is -0.98 nC/m.

• Are there any special ethical concerns arising from the use of animals or human subjects?

No

Reviewer #3 (Remarks to the Author):

This work describes the use of solid-state nanopores as sensors to detect hyaluronan. I enjoyed reading this manuscript as I think probing of non-DNA medically relevant biopolymers is an area where SS-nanopores could excel over other approaches especially at higher MWs. The work was thorough and the manuscript was well written. I don't think any major revisions are required and I recommend the work be published with some minor additions. I have some minor questions/comments listed below. I don't think the authors need to do additional experiments for this paper, some of the following are just things the authors should think about.

1) It would be nice to see dwell time histograms for the quasi-monodisperse HA samples. Do these have a long tail? The authors mention that clogging was only a major issue at high concentrations and high voltage. Do the authors see long duration events for the low MW samples, indicative of sticking?

We have added event duration histograms (Figure S3) in response to reviewer comments above. While the observed populations are not as defined as the ECD histograms (due predominantly to molecular folding, etc.), we did not observe a significant number of very long events for any population that would be indicative of sticking, as suggested.

2) At the higher event rates, for the longer dwell time events, what is the Poisson probability of two molecules translocating at the same time? In a polydisperse sample co-translocation of a high MW HA with a low MW HA would look just like a slightly higher MW HA (with a fold).

Considering even the highest event rate ($\sim 90 \text{ s}^{-1}$) and the longest duration events ($\sim 1 \text{ ms}$) measured in our work, the Poisson probability of two molecules passing through the pore simultaneously is 0.4%. While the reviewer is correct that such events would be misinterpreted, the chances of them occurring are vanishingly small.

3) How reproducible are the ECD distributions from pore to pore? If I measure a standard curve on one pore, how well would it work on another? The authors suggested that once measured the standard curve could be used for subsequent measurements, but it would be nice to have a bit more to back this up or clarify their statements.

We have included a new plot of ECD vs MW measured at 200 mV (Fig. S9) that includes the data shown in the manuscript along with data obtained from 6 additional nanopore devices ranging from 6.5 to 8.6 nm in diameter. These data demonstrate remarkable consistency and suggest that the standard curve can be applied broadly.

4) Most people in the nanopore field are used to thinking about DNA as it is the most common polymer so adding some of the polymer properties of hyaluronan would be nice (if known), like linear charge density, contour length or end-to-end distance vs MW, persistence length,... The amount of folding and translocation dynamics will depend on these properties. A quick search gives a persistence length of $\sim 5 \text{ nm}$ coupled with the very high ionic strength, should I be thinking of HA as a highly compact blob in the LiCl buffer? It would be nice to see a couple of events zoomed in, perhaps for the high MW HA. Do these look like dsDNA events with folding (given the quantized current histograms, as the authors point out) or are they more like ssDNA events? Does the HA look like it's captured on end or along the molecule?

We have added charge density, isoelectric point, and the length of a disaccharide unit (which, when combined with the MW range already provided, indicates contour length range). Typical translocation events have been added as Fig. S2 in response to Rev 2 comments above. We note that persistence length has been studied by other groups (see doi: 10.1529/biophysj.104.049361 and 10.1016/0003-9861(77)90008-X) and found to be in the single nm range. However, persistence length is highly dependent on solvent conditions and the results from previous studies (in very low ionic strengths) cannot be applied easily to our high-salt measurement buffer. Consequently, we do not provide the value in the manuscript.

5) The author's briefly allude at bandwidth issues. How is their SNR at 10kHz?

To address this concern, we measured the RMS noise for baseline current across a series of filter frequencies and then used the constant unfolded HA current level to determine the SNR for each. Results are shown in Fig. R2 and indicate that future iterations of the work may be able to use higher frequencies with the same 5 σ threshold level to improve resolution.

Figure R2. SNR dependence on filter frequency
Empirically derived signal-to-noise ratio as a function of low-pass filter frequency for a typical SS-nanopore.

Not for this manuscript, but as they do future work I would urge the authors to consider:

i) Does the bead immunoprecipitation introduce selection bias in the polymer length distribution? There are several ways to check this, one could be mixing quasi-monodisperse HA samples and measuring them before and after bead immunoprecipitation.

The HA-binding protein used in our study, versican, is commonly used in ELISAs where it has shown no significant size selectivity in the range on which we have focused. However, we are in the process of verifying this experimentally using the approach the reviewer described. Thank you for this suggestion for our future studies.

ii) In my experience with LiCl buffers >4M LiCl, we would often see spurious translocation events in the blank buffer control, which we attributed to nano-size salt precipitate, even though the solubility of LiCl is in the high 15M range. These would be visible under DLS measurements. I would urge the authors to be on the lookout for these when using very high molarity LiCl buffers.

Spurious translocation events of this kind would be prominent in the nanopore ionic current measurements even without HA (or other molecules) added into the measurement buffer. We do not observe such events over significant measurement times (c.f. Fig. 1b, -200 and -400 mV). As a matter of course, our buffers are *(i)* treated with an ultrasonic bath for mixing and *(ii)* passed through a 0.45 μm filter prior to use. These precautions may have contributed to the absence of such spurious signals, and so we have added this information to Materials and Methods.

Reviewers' Comments:

Reviewer #1:

Remarks to the Author:

The authors answered and took into account all my questions/comments. The manuscript was already well written and clear to read and the few points that were a bit unclear to me are now clearly explained.

I am in favor of publication of this work.

Reviewer #2:

Remarks to the Author:

The points raised in the previous round of reviews have been addressed and I am pleased to accept the manuscript for publication.

Reviewer #3:

Remarks to the Author:

The authors have adequately addressed all of the comments. I support publication of this manuscript.